# Towards a Secure and Sustainable Internet of Medical Things (IoMT): Requirements, Design Challenges, Security Techniques, and Future Trends

Bharat Bhushan [1,*], Avinash Kumar [2], Ambuj Kumar Agarwal [1], Amit Kumar [3], Pronaya Bhattacharya [4] and Arun Kumar [5]

1    Department of Computer Science and Engineering, School of Engineering and Technology, Sharda University, Greater Noida 201310, Uttar Pradesh, India
2    SITAICS, Rashtriya Raksha University, Lavad 382305, Gujarat, India
3    Department of Computer Science and Applications, School of Engineering and technology, Sharda University, Greater Noida, 201310, India
4    Department of Computer Science and Engineering, Amity School of Engineering and Technology, Amity University, Kolkata 700135, West Bengal, India
5    Galgotias College of Engineering and Technology, Greater Noida 201310, Uttar Pradesh, India
*    Correspondence: bharat_bhushan1989@yahoo.com; Tel.: +91-995-897-3930

**Abstract:** Recent advances in machine-to-machine (M2M) communications, mini-hardware manufacturing, and micro computing have led to the development of the Internet of Things (IoT). The IoT is integrated with medical devices in order to enable better treatment, cost-effective medical solutions, improved patient monitoring, and enhanced personalized healthcare. This has led to the development of more complex and heterogeneous Internet of Medical Things (IoMT) systems that have their own operating systems and protocols. Even though such pervasive and low-cost sensing devices can bring about enormous changes in the healthcare sector, these are prone to numerous security and privacy issues. Security is thus a major challenge in these critical systems, one that inhibits their widespread adoption. However, significant inroads have been made by the on-going research, which powers the IoMT applications by incorporating prevalent security measures. In this regard, this paper highlights the significance of implementing key security measures, and essential aspects of the IoMT that make it useful for interconnecting various internal and external working domains of healthcare. This paper presents state-of-the-art techniques for securing IoMT systems, in terms of data transmission, collection, and storage. Furthermore, the paper also explores various security requirements, inherent design challenges, and various security techniques that could make the IoMT more secure and sustainable. Finally, the paper gives a panoramic view of the current status of research in the field and outlines some future research directions in this area.

**Keywords:** Internet of Medical Things (IoMT); security; privacy; healthcare; medical devices; cyber security





## 1. Introduction

The Internet of Medical Things (IoMT) has emerged from the existing Internet of Things (IoT) technology, specifically dealing with the healthcare of patients. Similar to the IoT, the IoMT is also interconnected to various devices for communication with each other. The communication could be wireless, wired, or both, depending upon the devices and infrastructure developed. Smart cities are emerging at a rapid pace, and hence the need for the IoMT is also increasing [1]. The IoMT has the potential to reduce expenses incurred in the healthcare sector [2]; the sector is predicted to make a saving of USD 300 billion through the use of IoMT-based systems [3]. The revenue that was generated by IoMT-based systems in the year 2017 was USD 28 billion, and it is expected to grow up to USD 135 billion [4]. This will lead to a huge increase in the number of investors in IoMT-based systems.

Security, in any system, is a must, as it not only protects the client's data and information, but also protects the reputation of the organization [5]. Therefore, the IoMT needs to be protected from various types of cyber-attacks. The probability of cyber-attacks increases in the case of the IoMT because it works on the concept of the IoT, in which devices, protocols, and operating systems all distinctly constitute the heterogeneous environment [6]. The heterogeneity makes the system more vulnerable to cyber-attacks [7,8]. The main motivation for an attack on the IoMT is the value of the data stored within it, for the monitoring and treatment of patients. The average cost of IoMT data is estimated to be 50 times higher than in other sectors. This high value of IoMT data creates a greater risk of cyber-attack. Moreover, the devices used in IoMT-based system, such as closed-circuit television (CCTV) cameras, do not have patch update facilities, and thus require replacement if a vulnerability is found.

Considering the above two points, securing the IoMT is a necessity. Various techniques are used for achieving security and preserving privacy in IoMT-based systems. Symmetric key cryptography could be used as one of the solutions for preserving security in the IoMT [9]. The asymmetric key cryptography concept could also be used for protecting against data leakage in IoMT-based systems [10]. Apart from these two key-based techniques, key-less techniques could also be used for dealing with security and privacy issues in the IoMT [11]. These key-less techniques include token-based security, biometric-based security, proxy-based security, and blockchain-based security [7,12]. These security concepts are very useful as they are particularly sensitive to anomalies in the system. This anomaly detection helps the IoMT system, not only in detecting previously occurred cyber-attacks, but also in detecting new ones [13–16].

Limaye et al. [17] introduced innovative microarchitectures and suggested improvements that will allow for the efficient execution of future IoMT applications. Pritam et al. [18] proposed a comprehensive taxonomic review of current IoT-based sensor systems developed by the IoT market cap for taxonomic representations. Their research highlights the security and privacy challenges related to sensor data, and future strategies for enhancing the existing security vulnerabilities. Al-Turjman et al. [19] provided an overview of the IoMT, emphasizing future development, research objectives, and related applications. Authors presented a generalized IoMT structure that includes the three elements of data gathering, communication gateways, and servers/cloud. Sun et al. [20] aimed to efficiently use high quality healthcare resources, while working within the constraints of the present medical environment and medical-related equipment, to process and evaluate medical big data in a timely manner. They also concentrated on the improvements that cloud computing, edge computing, and artificial intelligence technologies have brought to the IoMT. In another work, Guangjun et al. [21] showed a triple topic intent-based security control (TS-PBAC) that is suitable for blockchain-enabled reliable transaction networks. Authors lay out an individual-centric security and confidentiality mechanism for access control, with various purposes and roles in IoMT scenarios. Ghubaish et al. [2] demonstrated cutting-edge methods for securing information in IoMT systems. The work provides a detailed analysis of potential physical and network threats that could endanger IoMT systems. The majority of security precautions do not take various forms of attacks into consideration. Ashfaq et al. [22] presented an extensive analysis of the numerous studies carried out to develop and improve the IoMT. They also included in-depth analysis of the benefits and downfalls of the various communication technologies now in use. In another work, Awad et al. [23] presented an in-depth analysis of an MEC (mobile edge computing)-based IoMT healthcare system. However, although the IoMT has been widely applied over the last 3 years, not enough comprehensive studies shed light on its security and privacy aspects. In contrast to the works discussed above, this paper brings forth a holistic approach for IoMT applications, highlighting motive, security requirements, concerns, and future research directions. In summary, the major contributions of this paper are as follows.

- This work presents the background of the IoMT and the motives for its wide acceptance to build the foundation for an understanding of the heterogeneous features of the IoMT.
- This work presents the various parameters that make the IoMT vulnerable to cyber-attacks.
- This work scrutinizes various security and privacy requirements in IoMT-based systems.
- This work discusses the major design challenges in an IoMT environment, and outlines various techniques used for resolving such security issues.
- This work presents a state-of-the-art solution using various methods to make the IoMT safer for application with humans.
- Finally, this work outlines several open research challenges that can help future researchers working in this emerging research area.

The remainder of the paper is organized as follows: Section 2 represents the background of the IoMT. Section 3 deals with the security requirements and design challenges of the IoMT. Section 4 presents security techniques in the IoMT. Section 5 presents a taxonomy of security protocols in the IoMT. Finally, Section 7 presents the future research directions, followed by a conclusion in Section 8.

## 2. Background of the IoMT

The rise in interconnected devices for human activities has led to a burgeoning exchange of data. The Internet of Things (IoT) has become a platform for various systems, including the Internet of Medical Things (IoMT), which is one of the most important areas in terms of human benefit [24]. The IoMT is useful for interconnecting medical devices and applications associated with them [2]. The IoMT is particularly useful for the remote monitoring of patients and the analysis of medical information [25]. Real-life data analysis is vital when gathering medical data in order to prevent a sudden attack in patients. The IoMT has increased the life expectancy of humans by monitoring patients' health at consistent times. The below sub-sections contain an in-depth discussion of the IoMT.

### 2.1. Defining the IoMT

Systems based on the IoMT concept typically have three levels, namely, the sensor, the personal server, and the server. Figure 1 presents the architectural view of this three-layer IoMT-based system. Importantly, most of the recently proposed work uses this architecture [26]. Sensors, along with other medical devices, are placed at the sensor level; these build up the local network and are termed the body sensor network (BSN) [27]. To provide services such as low powered Bluetooth communication (LPBC), and radio frequency identification (RFID), as well as near field communication (NFC), layers such as the sensor and personal server levels are used. LPBC is particularly useful, as it could be implemented in both star as well as mesh topologies. Those devices which are implantable could be improved using NFC and RFID. NFC and RFID are useful in device-to-device communication within a short distance. The physiological data and information are vital, and these are gathered using medical devices and transmitted to personal servers. The personal servers consist of on-body and off-body devices. The on-body devices include smart phones, tablets, etc., and off-body devices include routers, gateways, etc. These personal servers are useful in processing as well as storing the data at a local level before transferring it to the centrally located medical server (MS). The personal server must operate without any issues when connectivity to the MS is lost because of network issues. Various healthcare systems based on the concept of the IoMT have been proposed, among which BSN-Care is one of the most recent [28].

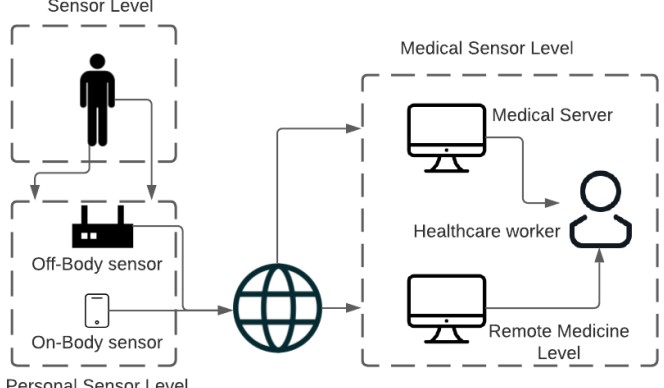

**Figure 1.** IoMT architecture.

### 2.2. Motives for IoMT Acceptance

Acceptance of the IoMT is essential to its application and use in the field of technology. The points below cover the important rationale that led to the adoption of the IoMT, as stated by Gus Vlahos [29].

- The use of the IoMT enhances the quality of the clinic. The Memorial Hermann Health System, located in Texas, in the United States, has adopted the IoMT for various activities, such as sending messages, scanning barcodes, and transmitting images.
- The perception of the generally functional and efficient automation of the IoMT helps in empowering efficient connectivity. One example could be the use of smart pills that can send messages, as well as alert signals, to the doctors who are associated with patient monitoring.
- Practical implementations of the IoMT can also be found in remote monitoring, which is performed via gathering data and transferring it to the relevant analyst, which could assist them in managing the patient's illness before it becomes more complicated. A practical example could be seen in the form of UCLA Health and Children's Health, located in Dallas.
- The IoMT also helps in conserving bodily health by accumulating and sending a person's health data to their healthcare practitioner. Practical implementation of this could be seen in the Apple Watch 6, which provides the user with alerts regarding the presence of oxygen in their blood.

### 2.3. IoMT Types

The IoMT is essential for the improvement of services relevant to various medical conditions. Devices such as pacemakers are implantable devices, whose performance can be enhanced greatly. The IoMT is broadly categorized into two types, Implantable Medical Devices (IMD) and the Internet of Wearable Devices (IoWD). These are thoroughly discussed in the below points.

- IMD refers to those devices which could be used to replace, support, or enhance the biological structure. One practical implementation could be seen in controlling the abnormal rhythm of the human heart using a pacemaker. The pacemaker supports the body by maintaining a consistent heartbeat in the case of an increase or decrease in heart rate from the normal human range [30]. A pacemaker will last longer if its power consumption is less; typically, they tend to last from 5 to 15 years, approximately [31].
- An example of the IoWD is typically worn by individuals to monitor their biometric data, such as heart rate, which could help to enhance their overall health. Devices such as blood pressure monitors (BPM), electrocardiogram monitors (ECG), smartwatches, etc., are examples of the IoWD [32]. Nutrition has become one of the major concerns for humans, and noncritical patients are widely monitored using fall detection and ECG readers [33,34].

### 2.4. Sensors for the IoMT

The IoMT-based application relies on the filed sensor and other sensors to carry out their designated task as per the IoMT architecture. The sensors used in the system are the most vital elements, as they are the first receptors for IoMT-based applications. In the case of the IoMT, there are various categories of e-healthcare sensors, such as disposable e-health sensors, connected e-health sensors, IoT-market cap sensors, and miscellaneous sensors.

Disposable e-health sensors that can respond to temperature, pressure, integral signals, etc., come under this category. MAX30205 and MLX90614 sensors fall under this category [35]. Connected e-health sensors of the IoMT should be connected at all times to ensure consistency in the communication transmitted over the network [36]. The pressure, position, temperature, etc., are the essential attributes measured by this sensor in the IoMT. IoT-Market cap sensors have flourished in the market. Smart Thermometer (ST) and Kardia Heart (KH) monitoring systems are some of the popular home-based health monitoring systems. Wearable fitness sensors are abundant on e-commerce websites. Miscellaneous sensors play a vital role in the life of pregnant women. The Internet of DNA (IoDNA) is another milestone achievement for genome mapping used for advanced medication systems. Synthesis of genetics using smart DNA (SDNA) could be utilized taking DNA measurements that could be uploaded over IoT-genome DNA for analysis. This concept is also used for predicting genetic defects in newborns. Figure 2 presents the classification of sensors. Table 1 presents the comparison of various sensors in terms of their performance and usage.

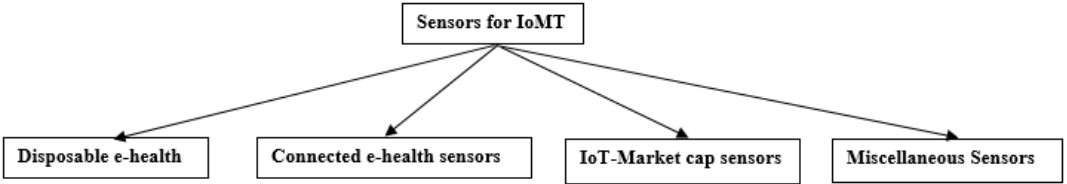

**Figure 2.** Types of sensors in the IoMT.

**Table 1.** Sensors and their parameters.

| S. No | Product | Type of Sensor | Reference | Support Priority | Disease/Monitoring | Cost | Data Usability | Energy Consumption |
|---|---|---|---|---|---|---|---|---|
| 1 | Proteus Digital Monitor | Clinical biometric | https://www.proteus.com/, accessed on 10 January 2023 | Yes | Hypertension, diabetes | Very high | Average | Very high |
| 2 | Obaa | Clinical | https://www.obaawoman.com/, accessed on 10 January 2023 | Yes | Patient waiting time reduction | Low | High | Average |
| 3 | OMsignal | Brain and fitness | http://omsignal.com/, accessed on 10 January 2023 | Yes | Wellness care | High | Average | High |
| 4 | Thalmic Labs | Home monitoring | https://www.bynorth.com/, accessed on 10 January 2023 | Yes | Virtual reality of health status | High | High | High |
| 5 | BabyBe | Sleep, infant and woman care | http://www.babybemedical.com/, accessed on 10 January 2023 | Yes | Bio signal between mother and premature infant | High | Average | Low |
| 6 | AdhereTech | Clinical | https://www.adheretech.com/, accessed on 10 January 2023 | Yes | Regular medication | Low | Average | Average |
| 7 | Pacifier | Sleep, infant and woman care | https://bluemaestro.com/, accessed on 10 January 2023 | Yes | Body temperature | High | High | High |
| 8 | CYCORE | Clinical biometric | http://cycore.ucsd.edu/, accessed on 10 January 2023 | Yes | Cancer | Average | High | Very high |
| 9 | Zeeq | Sleep, infant and woman care | https://sleeptrackers.io/zeeq-smart-pillow/, accessed on 10 January 2023 | Yes | Sleep | Average | Low | Average |
| 10 | Halo Neuroscience | Brain and fitness | https://www.haloneuro.com/, accessed on 10 January 2023 | Yes | Cognitive task management | Average | Low | Average |
| 11 | Voluntis | Clinical | https://www.voluntis.com/, accessed on 10 January 2023 | Yes | Cancer self management | Very high | Average | Very high |



**Table 1.** *Cont.*

| S. no | Product | Type of Sensor | Reference | Support Priority | Disease/Monitoring | Cost | Data Usability | Energy Consumption |
|---|---|---|---|---|---|---|---|---|
| 12 | TuringSense | Home monitoring | https://www.turingsense.com/, accessed on 10 January 2023 | Yes | Rehabilitation, posture correction, virtual reality | Average | Low | High |
| 13 | Quantus | Clinical biometric | https://quanttus.com/, accessed on 10 January 2023 | Yes | Sleep, diabetes, blood pressure | Low | Low | Low |
| 14 | Triggerish | Brain and fitness | https://www.sensimed.ch/sensimedtriggerfish/, accessed on 10 January 2023 | Yes | Irregular fitness tracking | Low | Average | Average |
| 15 | Teletracking | Clinical | https://www.teletracking.com/, accessed on 10 January 2023 | Yes | Patient–doctor communication | Average | High | High |
| 16 | Cue Health | Home monitoring | https://www.cuehealth.com/, accessed on 10 January 2023 | Yes | Inflammation, influenza, fertility, testosterone | Low | High | Low |
| 17 | Biostrap | Brain and fitness | https://biostrap.com/, accessed on 10 January 2023 | Yes | Sleep recovery and performance management | Low | High | Low |
| 18 | Sotera Wireless | Clinical biometric | http://storeawireless.com/, accessed on 10 January 2023 | Yes | Blood pressure, fall detection | Average | Very high | Low |
| 19 | Beddit | Sleep, infant and woman care | https://www.beddit.com/, accessed on 10 January 2023 | Yes | Sleep and wellness | Average | High | Low |
| 20 | BioSerenity | Home monitoring | https://www.bioserenity.com/, accessed on 10 January 2023 | Yes | Epilepsy monitoring | Very high | Average | Average |
| 21 | MC10 | Clinical biometric | http://mc10inc.com/, accessed on 10 January 2023 | Yes | Sleep, posture, heart rate | Low | Low | Low |
| 22 | Ovia | Sleep, infant and woman care | https://www.oviahealth.com/, accessed on 10 January 2023 | Yes | Ovulation | Low | High | Average |
| 23 | Breezhaler | Home monitoring | https://www.medicines.org.uk/emc/product/3496/smpc, accessed on 10 January 2023 | Yes | Asthma | Average | Low | Average |
| 24 | NeuroSky | Brain and fitness | http://neurosky.com/, accessed on 10 January 2023 | Yes | Mental and physical integration | Low | Average | Low |
| 25 | Evermind | Clinical | http://evermind.us/, accessed on 10 January 2023 | Yes | Daily activity | High | Low | High |

### 2.5. State-of-the-Art Strategy for Telesurgery or Remote Surgery

Tele-surgery has emerged as a possibility with the advancement of communication linked to the transition from 4G to 5G. This has made possible remote treatment in societies where the movement of specialists is complex. The 5G-enabled tactile internet (TI)-based tele-surgery system operates over a network, for Healthcare 4.0-based state-of-the-art applications [37]. The tele-surgery system works using the principle of a master–slave system, consisting of an operation site robot, which is responsible for performing the desired surgery task based on the command and control provided by the human system interface. the system works in concert with the haptic device, earphones for receiving audio input commands, and a video console. Therefore, the need for consistent connectivity is particularly crucial for any IoMT-based infra-structure where there is a use of remote connection to tackle medical emergencies in case of the unviability of medical services in that particular geographical location. Additionally, in the age of cyber war among nations, there are possibilities that a nation's critical assets, which may include healthcare, will be targeted and thus it is highly important to safeguard against jamming or channel-based attacks. Additionally, all remote or tele-surgery activities involve the use of a command and control server (C2 server), which is one of area of great interest for hackers. Such systems are vulnerable to distributed denial of service (DDoS) attacks, which may include ping of death (PoD), UDP flooding, IP address spoofing, SYN flood, etc., and any of these attacks, performed on the C2 servers could lead to catastrophic loss of human life. Therefore, in order to understand the threats presented by the use of remote connection, it is vital to investigate and dive deep into the important medical services carried out remotely.

### 2.5.1. Teleoperation

This method of tele-surgery is based upon the concept whereby the master or controller transmits position commands, and the salve manipulator is responsible for adhering to the command. Apart from the server and processor, the system is comprised of the master manipulator (MM), salve manipulator (SM), and servo system (SS) [38,39]. The MM is responsible for controlling the robot, whereas the SS and SM are responsible for operating the robot. The SS tackles the output-controlled data of the SM. Software is used to analyze the bulk of the analog signals (AS), and also constructs the position coordinates.

### 2.5.2. Endoscopic Telesurgery

This tele-surgery technique, in comparison to earlier methods of tele-surgery, is more evolved. Laparoscopic cholecystectomy is considered to be the first robot-assisted surgery. Later, the Da Vinci surgical robot emerged and was considered to be extremely efficient for short-distance remote surgery. Despite the short range of the Da Vinci surgical robot, it could be used over larger distances, by using procedure control (PC) as well as including the remote guidance system (RGS). The Raven II System, as well as the Lapabot System, are two more widely used robots in tele-surgery [40,41].

### 2.5.3. Neurosurgical Telesurgery

This deals specifically with remote surgery on neurosystems. The Socrates Robot Remote Cooperative System was the first system developed that was mainly responsible for surgical training [42]. Neuro-Arm is another tele-neurosurgery-based system that is useful in fusing MRI data with a 3D view, but it is also useful in short-distance tele-neurosurgery [43]. China has developed tele-robotic-directed neurosurgery, which is one of its kind and is capable of all fourth-generation tele-surgery features [43].

### 2.5.4. Orthopedic Telesurgery

In this method of tele-surgery, there are two main components involved, namely a high-end 3D camera, and a robot responsible for tackling the master–slave-based communication. The surgeon performs the task, first by visualizing the patient, by looking at the screen, and then, in order to control the remote robot, the surgeon uses a haptic arm (HA). In response to this, the robot arms interpret the instructions given by the surgeon, who is sitting remotely. This arrangement is useful for treatment, as the robot mimics the natural movement of the surgeon. The main factor enabling this transition of input from surgeon to robot is haptic technology (HT). HT has become more efficient with the inclusion of artificial intelligence and virtual reality [44].

## 3. Security Requirements and Design Challenges

Security is a vital issue for data exchange between medical devices in IoMT-based systems. The data of the patient are very sensitive, as they are particularly personal. The medical history and symptoms of the patients are processed and recorded in IoMT-based systems. Therefore, this data must be protected from the attacker. In order to better understand the various security parameters [45–48], the four most important are discussed in the below enumeration.

### 3.1. Security Requirements

- Confidentiality/Privacy: Confidentiality, or privacy, is the top priority, as a huge amount of sensitive and personal data is processed and stored across IoMT devices. These data should be accessible to the authorized user via a proper authentication mechanism; furthermore, the stored data should be encrypted to avoid ease of access by an adversary. The encryption adopted must be secure enough to safeguard from attackers [46].
- Integrity: The integrity of data in the IoMT is essential, as these inputs are used for the treatment of the patients. Integrity ensures that the data has not been modified,

either during transmission or during the storage process. The modification of data may consist of deleting it, adding false values, etc. It is important to safeguard the sensitive data of the IoMT to stop unauthorized access.

- Authentication: The validation of authorized users for communication is key to performing identity authentication. To authenticate an identity, both communicating parties must mutually verify themselves. The transfer of data and information occurs after mutual authentication. The IoMT consists of various services, including the cloud, that need adequate authentication. The authentication mechanism may vary according to the various IoMT-based applications [47].
- Non-Repudiation: This is particularly crucial because an illegal entity could not deny the validity of the messages. To validate the messages, the proof of origin is mentioned along with the integrity of the data. The denying of the message becomes extremely tough when the source or origin is mentioned. The concept of a digital signature is widely used for implementing non-repudiation.
- Availability: This feature ensures that the information and services are accessible to authorized users only. The availability feature is exploited by the adversary or attacker by executing a denial-of-service (DoS) attack. This attack is generally launched when confidentiality and integrity of the system remain intact and the attacker is unable to compromise these two features [48].
- Backward and Forward Secrecy: The backward and forward features are an integral part of the IoMT-based system since it consists of hardware devices in large numbers. Forward secrecy suggests that, if any device leaves the IoMT system, then it should be discontinued, so that it could not access any communication within the existing system. Moreover, in case of backward secrecy, newly installed devices in IoMT systems should not have any access to previously transmitted messages.

### 3.2. Design Challenges in the IoMT

- Postural body movement: The sensors which are used in on-body medical devices, as well as other sensors, are usually placed in a group. The movement of the patients using these devices and sensors is not consistent, as they are highly mobile. The transmission used to monitor postural body movement could be optimized by a quality change associated with the movement of the patient [49].
- Temperature rise: A temperature rise is generally observed in any hardware-based system. In the case of the IoMT, two main factors raise the temperature of the system. Radiation through the antenna is the first cause, while the consumption of power is the other major cause [50].
- Energy efficiency: Energy efficiency is preserved in IoMT-based systems by designing them in such a way as to make optimal use of energy on local devices or sensor nodes, and also optimize the energy consumption of the overall network across its lifetime. This implementation is especially important in the case of surgical devices in IoMT-based systems, where the battery is the main source of energy [51].
- Transmission range: The transmission, when it occurs across a very short range along with movement of the body, sometimes leads to disconnection, as well as re-partitioning in the sensor present in the IoMT. There is a need to minimize the total number of sensors on the patient's body to reduce disconnection. IBS is one of the methods whereby transmission is made more optimal [52].
- Heterogeneous environment: IoMT-based systems are generally comprised of various devices and sensors, which are manufactured by different manufacturing companies. These devices use different architectures for their operation. Thus, the system becomes highly heterogeneous. Therefore, the network must be capable enough to tackle these heterogeneities to route the data and information properly in the IoMT-based system.

### 3.3. Concerns in the IoMT

Security concerns are high in IoMT-based systems as the system is heterogeneous, and also consists of wireless communication, thus making it vulnerable to wireless network-based attacks. The eavesdropping and man-in-the-middle (MitM) attacks are two main major attacks that are possible against wireless-based communication. Privacy concerns are high in the case of the IoMT, and attacks on network traffic aiming to gain sensitive information about patients are a serious threat to IoMT-based systems. Passive attacks are more prevalent in systems where there is a possibility of identity theft. Trust concern relates to trust management, and deals with the accountability of the service provider to the users. Any breach of sensitive and private data results in the loss of the trust of the user in the service providers. A lack of training among healthcare workers and employees could lead to permanent damage to the patients, and might also result in loss of life. Data falsification, or the transmission of wrong data, could result in delivering the wrong drugs, or the wrong amount of drugs, into a patient's body, which might create serious issues. Accuracy becomes paramount when robots are used in an IoMT-based system, as their mishandling could have fatal results.

### 3.4. Prevalent Attacks in the IoMT

Although there exist some state-of-the-art techniques to secure IoMT systems, nowadays the adversaries are also equipped with advanced techniques to exploit them. There are various aspects through which the IoMT can be targeted by cyber-attacks [53]. In general, adversaries try to compromise the confidentiality–integrity–availability (CIA) triad of an IoMT network. Thus, their attacks can be either active or passive, and are categorized as attacks against data confidentiality, privacy, device authentication, and many more [54]. The Sybil attack is also a prominent tool used to target IoMT networks [55]. Table 2 presents various cyber-attacks prevalent against the IoMT.

**Table 2.** Cyber-attacks in the IoMT.

| Category | Attacks | Possible in IoMT |
|---|---|---|
| Data confidentiality attacks [2,56] | Man-in-the-middle (MitM) Packet sniffing | Yes |
| Social engineering attacks [24] | Pretexting Baiting attack | Yes |
| Privacy attacks [57] | Black-box attack White-box attack | Yes |
| Availability attacks [58] | Distributed DoS (DDoS) Flooding attacks | Yes |
| User or device authentication attacks [59] | Brute forcing, masquerading, replay attacks, session hijacking, rainbow attacks, dictionary | Yes |
| Malware attacks [60] | Spyware, Trojan, rootkit | Yes |

### 3.5. Existing Security Framework for IoMT-Based Applications

Security of the IoMT is crucial, as this type of infrastructure consists of particularly sensitive and private data that ranges from the personal health condition of a person to his/her insurance details. According to the Health Insurance Portability and Accountability Act of 1996 (HIPAA), it is necessary for any organization dealing with the medical data of a user to safeguard the data, and only give privilege for the electronic transfer of data to a third party for processing when the user has given their consent for the same [61]. Therefore, if the IoMT system includes data from a US Citizen, this Act needs to be followed by the organization. Additionally, if the IoMT infrastructure falls within the European Union, the IoMT-based organization must then abide by the General Data Protection Regulation (GDPR) [62].

J. Rauscher et al. [63] proposed a framework model on security for the IoMT, known as the IoT safety and security architecture analysis framework (IoT- S2A2F). This model is useful in analyzing existing IoMT infrastructure, as well potential new infrastructure, for vulnerability assessment in order to tackle threat vectors. Additionally, models such as the architecture analysis and design language (AADL) are very crucial for threat hunting and finding vulnerabilities in a system based on the IoT infrastructure [64]. The misbehavior detection system (MDS) is yet another important model that deals with the privacy, as well as security, of the IoMT system. S. Rahmadika et al. [65] proposed a blockchain-based framework for enhancing the intrusion detection capability of blockchain. This model is especially effective when the system is very heterogeneous, as the blockchain can detect even slight changes in the value of the asset, and thus making the model very sensitive to intrusion upon the system. The security of the IoMT is critical because various devices of different architecture and protocols communicate with each other within it; therefore, it is important to segregate the network into layers. The use of the layered security model (LSM) would be helpful for pen-testers investigating points of weakness in IoMT-based applications. The safeguard of MDS is very crucial in avoiding harmful injection attacks, as a harmful SQL injection could lead to the complete credentials of patents being compromised. Choudhary et al. [66] have proposed a lightweight misbehavior detection scheme that relies on formal verification and automatic model checking in a medical cyber physical system. In another work, Astillo et al. [67] aimed to mitigate security threats, by using a specification-based misbehavior detection model, and validate the data integrity using outlier detection algorithms.

### 3.6. Risk Analysis and Threat Mapping

The risk analysis is a very crucial aspect of the security efforts, as it helps to analyze the level of threat. This analysis could be performed using the threat mapping technique, whereby the threats are mapped according to their severity. Table 3 depicts the threat and risk associated with the various IoMT elements.

**Table 3.** Threat and risk mapping.

| IoMT Assets | Possible Threat Entry | References | Risk | Severity |
|---|---|---|---|---|
| Gateway | Attack from WAN | Gao et al. [68] | The ARP table could be poisoned that is exiting in Gateway router | High: as it may reveal important IP of internal switches and routers |
| Helpdesk Workstation | LAN and WAN | Gopal et al. [69] | Virus could be inserted or phishing mail could be sent to reception. | Low: as generally this workstation does not consists of any permanent data, only appointment times and patient names |
| Web Server | WAN | Shah et al. [70] | Consist of the web application on which IoMT website and application would be running | High: this may lead to complete failure hospital management system and bring the organization back to pen and paper mode |
| MD | LAN and WAN | G. M et al. [71] | Consist of admin and other users' passwords | High: as it can compromise the complete digital infrastructure. |
| Filed Sensors | Hardware means | Elmahi et al. [72] | Jammer could be used to create noise | Moderate: this can reduce the efficiency of reporting of data to the centralized server performed by Filed Sensors |
| SIEM | WAN and LAN | - | If the monitoring framework itself becomes compromised then, then all internal and external attacks would not be visible | High: this will create problems in the monitoring of logs, networks and other vulnerable areas where attacks could take place. |

## 4. Security Techniques in IoMT

The above-explained attacks suggest that there is a strong need for securing the IoMT, as it could be vulnerable to cyber-attacks. The below subsections explain various vital security techniques required for securing the IoMT.

### 4.1. Symmetric Key Cryptography

Symmetric cryptographic algorithms are based on shared/secret keys between the communicating nodes, and require prior key generation. These algorithms are useful in IoMT systems for initiating secure connections and hierarchically accessing the patient's data. Furthermore, these facilitate two-factor authentication, in which other techniques, such as pattern-based and facial recognition, act as the second factor [73]. The role of symmetric key algorithms in IoMT systems is discussed in the subsections below.

#### 4.1.1. Hierarchical Access

This technique provides role-based authorization and facilitates hierarchical access to patients' data using a hierarchical role-based architecture. For instance, any nurse can administer medicines, but only an authenticated doctor can prescribe a new medication. The model requires a hierarchical security technique of relatively low complexity that enciphers the patient's personal information and deciphers only part of the data. Belkhouja et al. [74] proposed a role-based encryption standard that guarantees hierarchical access to patients' data and also overcomes the computational shortcomings of implantable medical devices (IMDs). Their standard achieves the desired encryption hierarchy by using the Chinese remainder theorem (CRT), where the users can access the patient's data according to the assigned privilege [75]. A more privileged user can access any data, whereas users with restricted privilege can access only the relevant data.

#### 4.1.2. Biometric Systems

Most IoMT systems rely on facial scanning for authenticating users in a continuous role-based authentication process. The shared keys serve as the first authentication factor, while facial recognition serves as the subsequent authentication factor. This enables a secure connection between the medical controller and the sensor [76]. Further, the system is capable of securing the overall medical setting and preventing the staff from having limited privilege to access the data. In another work, Belkhouja et al. [77] proposed an elliptic curve cryptography (ECC)-based authentication scheme that allows secure access to implanted devices and protects wireless key exchange. This works by using integrated fingerprint and electrocardiogram to create a lightweight, efficient, and secure authentication scheme for IMDs.

#### 4.1.3. Gait-Based Scheme

This scheme generates unique symmetric keys using the human walking pattern. Sun et al. [78] proposed a novel biometric cryptosystem that uses an artificial neural network (ANN) and gait signal energy variations for securing wireless communications in implantable and wearable healthcare devices. Their approach extracts similar features from the body sensors, and aims to generate on-demand binary keys without any manual intervention. The proposed scheme outperforms the existing state-of-the-art techniques in terms of the number of bits generated per gait cycle. A 128-bit key is generated to secure communications between the access points and IoMT sensors. Furthermore, the generation of binary keys at different times adds randomness to the keys without any direct system–user interaction.

#### 4.1.4. Cryptographic Hash Function (CHF)

CHF is a one-way function that maps an arbitrary size input to fixed-size data [79]. In healthcare scenarios, the initial parameters (such as a shared key and node ID) can be XORed together (to verify if one of the operands is different), and then hashed. The hash value is then shared with the gateway and sensor nodes to generate their keys [80]. IoMT systems can use authenticated key agreement protocols and secure communications by integrating the XOR operator, a symmetric key, and the CHF. Xu et al. [81] proposed a lightweight authentication technique that guarantees forward secrecy in wireless body area networks (WBANs) without exploiting asymmetric encryption standards. The proposed

scheme incurred low computational costs and is less vulnerable to security attacks, as compared to other lightweight security schemes. Alzahrani et al. [82] proposed a provably secure and reliable key agreement-based health monitoring protocol that is resistant to some of the most prominent attacks against WBANs, such as key compromised impersonation attacks and replay attacks.

### 4.2. Asymmetric Key Cryptography

Asymmetric cryptographic algorithms include two keys: a private key (for decryption/signature), and a public key (for encryption/validation). Everyone knows the public key, whereas only the owner knows the private key. Figure 3 depicts the process of encryption/decryption in asymmetric key cryptography. Some of the most widely accepted algorithms in this category include elliptic curve cryptography (ECC) and Rivest–Shamir–Adleman (RSA) [83]. ECC in particular is useful in securing IoMT systems, owing to its lightweight features. Furthermore, asymmetric keys can also provide two-factor authentication. Asymmetric keys can act as the first factor, and are supported by various techniques (such as smart cards used in hospitals) that serve as the second factor. The role of asymmetric key algorithms in IoMT systems is discussed in the subsections below.

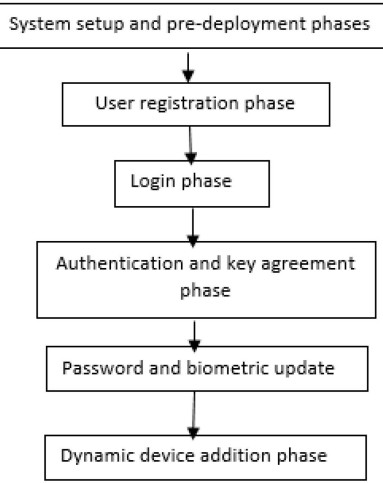

**Figure 3.** User device management in the IoMT.

### 4.2.1. Homomorphic Encryption (HE)

HE ensures data confidentiality and mitigates the mathematical complexity linked to data encryption. In IoMT settings, it maintains data privacy and allows only the patient to access their data by storing the encrypted data in the cloud layer. For example, IoMT sensors in smartwatches allow continuous data encryption and reveal the relevant data to medical staff only during an emergency, when necessary to perform a proper diagnosis. Sun et al. [84] fully leveraged HE to encipher patient data in a mobile healthcare network, in order to achieve computations on the cipher text. In another work, Jiang et al. [85] utilized HE to achieve privacy preservation in medical diagnosis. Farooqui et al. [86] proposed a human-centered model to identify and treat mental healthcare patients by integrating an encrypted cloud computing paradigm and emerging IoT wearable technologies. Guo et al. [87] proposed a homomorphic encryption-based privacy-preserving scheme that prevents the leakage of cluster center data and the disclosure of participants' privacy. Kara et al. [88] proposed a full HE scheme based on magic number fragmentation and twin key encryption. The proposed scheme is resistant to brute-force attacks and effective against cryptanalytic attacks in smart city applications.

### 4.2.2. CHF with ECC

Integration of ECC keys and CHF function can help in realizing a certificateless secure channel between medical doctors and their patients. This integration allows the sharing of

keys securely between the key generation server (KGS) and nodes in IoMT layers. Node ID, ECC public key, and other preliminary parameters are hashed with the help of CHF. The hashed value is then sent to the nodes in the IoMT gateway and sensor layers. Asymmetric keys can be generated from the received hash values, thereby solving the prominent issues of secret key sharing in symmetric cryptographic techniques [89]. Furthermore, this can mitigate the certificate management overhead for data sharing and storage in the cloud [90]. Considering the substantial increase in IoMT data sizes, it is advantageous to divide the medical data into subsets and share them securely using CHF and ECC keys. Maria et al. [91] integrated ECC and other software implementations to designing secure and privacy-preserving solutions for IoMT communications. The proposed scheme guarantees origin authentication, integrity, and data source anonymity, with complexity and computational requirements within acceptable limits. Zheng et al. [92] aimed to preserve data integrity using ECC and data encryption standards (DES) in electrocardiogram (ECG) monitoring systems. Ogundokun et al. [93] presented a crypto-stegno model to preserve the confidentiality of the patient's EHRs in an IoMT environment. Sowjanya et al. [94] designed a lightweight, robust, and anonymous ECC-based authentication protocol for securing medical data in WBANs.

### 4.2.3. Digital Signatures

Digital signatures have been employed for verifying data authenticity in a small IoMT system using the public and private keys of the sender for verification and signature respectively [95]. These can be integrated with the sensor's firmware in IoMT systems with an add-on to intercept and validate wireless communication [96]. This method mandates the storage of public keys of authorized users in the sensor's firmware for validation. Farahat et al. [97] aimed to guarantee secure authentication and data privacy using digital signatures and key rotation schemes. Kumar et al. [98] proposed a smart, escrow-free, identity-based cloud-centric IoMT system that gathers patients' data and outsources it to a medical cloud server after sign-encrypting and aggregating them.

### *4.3. Keyless Algorithm*

Security schemes based on blockchain technology, proxy-based techniques, and biometrics guarantee security without the need for pre-shared keys. The role of keyless algorithms in IoMT systems is discussed in the subsections below.

### 4.3.1. Blockchain Technology

Blockchain technology is employed in IoMT systems for security management and data sharing. However, due to strict communication requirements and larger data chunks, blockchain is susceptible to communication overhead, latency, and storage issues [99]. Nguyen et al. [100] designed a smart contract-based access control mechanism for securely sharing EHRs in IoMT systems. Similarly, Garg et al. [101] proposed a blockchain-enabled key agreement scheme aimed at providing secure key exchange between cloud servers, personal servers, and medical devices. In another work, Meng et al. [102] utilized blockchain for detecting malicious nodes by enhancing trust management in medical smartphone networks. Gao et al. [103] proposed an integration of blockchain technology and edge computing for safeguarding the confidentiality of data analysis in IoMT systems. The proposed framework relies on blockchain for authenticating the cloud service providers and IoMT devices in the network. Egala et al. [104] proposed a blockchain-based, decentralized, and automated access control technique for IoMT systems. The proposed technique utilizes a hybrid computing paradigm and solves the issues related to latency, high cost, and single point of failure. In another work, Jin et al. [105] presented a blockchain-based cross-cluster federated learning solution for secure data sharing in IoMT systems. Similarly, Awad et al. [106] integrated blockchain and edge computing to optimize the computational cost and latency of medical record sharing between entities.

### 4.3.2. Proxy-Based Systems

Proxy-based systems facilitate full duplex secure communication, and rely on a middleware device for controlling the communication between devices such as medical controllers and sensors. Kulac et al. [107] aimed to secure communications in implantable medical device (IMD) systems and provide full duplex secure transmissions using a protective security belt. Verma et al. [108] presented a provably secure and efficient proxy signature scheme for e-healthcare systems. In another work, Bhatia et al. [109] proposed a proxy-based, pairing-free, and lightweight re-encryption scheme for securely sharing the EHRs with the public cloud. In the proposed scheme, patients use the public key to encrypt data before outsourcing it to the cloud. Kulac et al. [110] aimed to protect IMDs from adversaries and achieve full duplex secure communications using a middleware device, named the protector jacket. The protector jacket uses spoofing-based beamforming techniques to support longer battery life and higher power efficiency. In another work, Li et al. [111] integrated the concept of public key encryption and proxy re-encryption to search healthcare records securely and flexibly.

### 4.3.3. Biometrics

One of the most widely accepted techniques to enhance security in IoMT systems is the use of biometric sensors. Patients and the medical staff can access the medical records only after biometric authentication in a medical setting. ECG-based sensors and fingerprints are the most commonly used biometrics. ECG-based sensors encrypt data based on heartbeat activities and fingerprint sensors rely on reading the fingerprint image. Fingerprint sensors are superior to ECG-based techniques in terms of computational overhead and message size. Zheng et al. [112] proposed a finger-to-heart authentication technique that aims to utilize the patient's fingerprint for granting access to the IMD. In another work, Zheng et al. [113] used fingerprint information to safeguard elderly people suffering from memory loss and protect them from malicious adversaries in IoMT systems. Shakil et al. [114] proposed a biometric-based authentication scheme to secure health data using a behavioral biometric signature.

The major security advances in IoMT scenarios are summarized in Table 4.

**Table 4.** Recent security advances in IoMT scenarios.

| Security Techniques in IoMT | References | Year | Major Contribution |
|---|---|---|---|
| Symmetric key cryptography | Liu et al. [73] | 2019 | Lightweight NTRU public key cryptography-based security protocol |
| | Belkhouja et al. [74] | 2019 | Role-based encryption standard to Overcome the computational shortcomings of IMDs |
| | Tutari et al. [76] | 2019 | Role-based authentication for IMDs |
| | Belkhouja et al. [77] | 2019 | Secure access to implanted devices and protects the wireless key exchange. |
| | Sun et al. [78] | 2019 | Biometric cryptosystems that use ANN and gait signal energy variations |
| | Xu et al. [81] proposed a | 2019 | Lightweight authentication technique to guarantee forward secrecy in WBANs |
| | Alzahrani et al. [82] | 2020 | Provably secure and reliable key agreement-based health monitoring protocol |

**Table 4.** *Cont.*

| Security Techniques in IoMT | References | Year | Major Contribution |
|---|---|---|---|
| Asymmetric key cryptography | Sun et al. [84] | 2017 | Fully homomorphic encryption in healthcare networks |
| | Jiang et al. [85] | 2019 | Privacy-preserving in IoT |
| | Farooqui et al. [86] | 2019 | Human-centered model to identify and treat mental healthcare patients |
| | Guo et al. [87] | 2020 | Homomorphic encryption to prevent leakage of patients' data |
| | Kara et al. [88] | 2021 | Magic number fragmentation and twin key encryption |
| | Kasyoka et al. [89] | 2020 | Pairing-free authentication protocol for WBANs |
| | Bhatia et al. [90] | 2020 | Healthcare data sharing using incremental proxy re-encryption |
| | Maria et al. [91] | 2020 | Secure and privacy-preserving solutions for IoMT communications |
| | Zheng et al. [92] | 2020 | Preserves data integrity in healthcare monitoring systems |
| | Ogundokun et al. [93] | 2021 | Preserves data confidentiality using crypto-stegno model. |
| | Sowjanya et al. [94] | 2021 | ECC-based authentication protocol for securing medical data in WBANs. |
| Keyless algorithm | Easttom et al. [96] | 2019 | Cyberthreats in implantable medical devices |
| | Farahat et al. [97] | 2018 | Secure authentication using digital signatures and key rotation schemes. |
| | Kumar et al. [98] | 2020 | Identity-based cloud-centric IoMT system |
| | Nguyen et al. [100] | 2019 | Smart contract-based access control mechanism for securely sharing EHRs |
| | Garg et al. [101] | 2020 | Blockchain-enabled key agreement scheme |
| | Meng et al. [102] | 2020 | Blockchain-based trust management in medical smartphone networks |
| | Gao et al. [103] | 2021 | Blockchain technology and edge computing to maintain confidentiality in IoMT systems. |
| | Egala et al. [104] | 2021 | Automated access control technique for IoMT systems |
| | Jin et al. [105] | 2021 | Blockchain-based cross-cluster federated learning solutions for secure data sharing in IoMT systems |
| | Awad et al. [106] | 2021 | Optimize computational cost and latency of medical record sharing |
| | Kulac et al. [107] | 2017 | Protective security belt to provide full duplex secure transmissions |
| | Verma et al. [108] | 2017 | Proxy signature scheme for e-healthcare systems |
| | Bhatia et al. [109] | 2018 | Lightweight re-encryption scheme for securely sharing the EHRs |
| | Kulac et al. [110] | 2019 | Protector jacket to protect IMDs from adversaries |
| | Li et al. [111] | 2020 | Public key encryption and proxy re-encryption for securely searching healthcare records |
| | Zheng et al. [112] | 2019 | Use the patient's fingerprint for granting access to the IMD |
| | Zheng et al. [113] | 2020 | Fingerprint information to safeguard elderly people suffering from memory loss |
| | Shakil et al. [114] | 2020 | Behavioral biometric signature to secure health data |

## 5. Taxonomy of Security Protocols in IoMT

The taxonomy discussed in this section focuses on various protocols that are vital for the IoMT. Moreover, the other function of security protocols in IoT communication environments, apart from providing security, is to store, as well as exchange, data. The optimality of security features is a crucial part, not only in the IoMT, but rather in every system which relies on channels of communication for their desired end task. Here, in the case of the IoMT, since the data are more sensitive (mental health of patient, drugs used by him or her) and more private (his relationship, as mentioned in insurance documents), this creates an area of interest for adversaries, and the weak protocol becomes a particularly vulnerable point for such a system. Protocol attacks, which are also sometimes referred to as

state-exhaustion attacks, are very dangerous. Therefore, it is important to use more secure protocols in the case of the IoMT. It is crucial to understand how the least privilege is being tackled using the access control (AC) concept and how the cryptography plays important role in making the protocols more resilient in detecting, preventing and mitigating state-exhaustion attacks.

Figure 4 shows the taxonomy of security protocols in the IoMT. The below subsections discuss the vital aspects of security protocols for the IoMT.

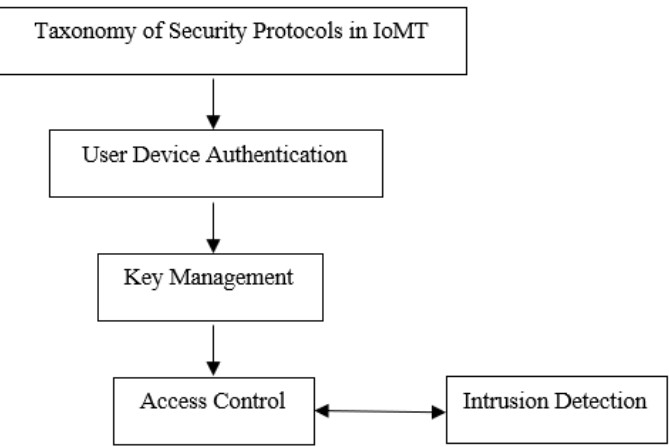

**Figure 4.** Taxonomy of security protocols in IoMT.

### 5.1. Key Management

The role of the key management (KM) protocol is to create, manage, and distribute cryptographic keys between the entities interacting with one another in the IoT/IoMT environment. Based on the requirements, the whole process is classified into various phases, such as key generation, key exchange, key usage, and key revocation. A "cryptographic technique" is followed by the key management mechanism that supplies the details of various static or mobile users, different IoT devices, as well as key servers. For secure communication, robust key management plays the most significant role in this context [115]. KM goes through four different phases. Figure 5 presents the various phases of key management in IoMT-based systems.

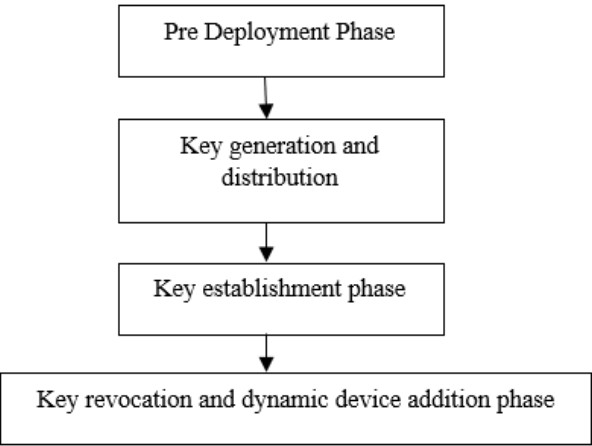

**Figure 5.** Key management in the IoMT.

### 5.2. User Device Authentication

The process of identifying and verifying the identities of the communication parties is known as user authentication. Often, in device or user authentication, the communicating parties carry out this verification among themselves before establishing the session key for

communication. This is termed mutual authentication [116]. Figure 3 explains the working scenario of user device management in IoMT-based systems.

### 5.3. Access Control

Permission management is crucial to avoid the mishandling of data and resources in any system. This is frequently implemented using an access control mechanism. Access control is intended to limit and manage access to resources that include data, devices, and networks. In the case of the IoMT, the devices are granted the privilege to access the resources. This is intended to improve the smooth functioning of the IoMT-based system. The longevity of the IoMT-based system could be increased by adding new devices, which could be generic or smart. This is required, as many devices are battery based and their batteries degrade with time. The physical attack could be performed when the adversary replaces the original device in the network with the infected one [117]. To secure such devices, the designing of IoMT-based systems should follow an access control mechanism [118–120]. The access control is implemented using node authentication (NA) and key establishment (KE). NA is essential in avoiding physical attacks upon the addition of a new device to the existing IoMT system, as it should first authenticate itself with the neighboring devices to confirm to them that it is an authentic device, not an illicit one. Moreover, KE helps to secure future communication across the whole system when a new device is added. This is achieved by enabling secret key sharing between the existing and newly added devices. Apart from these, it is very important to grant access to legitimate users only.

### 5.4. Intrusion Detection

Detection of intrusion in a system is vital for the normal function of the system. Intrusion detection system (IDS) is very helpful in the analysis of malicious activities that occur inside the network of the targeted system. The IDS is capable of detecting and preventing various types of attacks attempted against the system. The IDS, when installed in the IoMT system, monitors the network and detects any anomalies initiated in the traffic. If an anomaly is detected, IDS will then alert the system and corresponding devices will take appropriate action based on this alert. The IDS could send a message to the admin to block an IP that seems malicious [121]. The adversary can sometimes physically access the device and replace it with an infected one. Moreover, a power analysis attack could be used by the adversary to gain sensitive information from the system [122,123]. This could give credential information to the adversary and, using these gained credentials, the adversary could implant his own devices. These maliciously crafted devices could be used by an adversary to execute a more devastating attack. The IoMT could also be attacked by various types of viruses and malware. Moreover, new malware could also attack the existing IoMT system. Further details of the detection mechanism are discussed in the below enumerations.

## 6. Future Research Directions

The IoMT is increasing the efficiency and accuracy of the treatment of patients through real-time monitoring of patients' healthcare data. Despite the IoMT being an extremely recent and advanced technology, the system is vulnerable to cyber-attacks because of its heterogeneous nature. The below subsections present a few vital points to be incorporated into future research.

### 6.1. Scalability of Malware Detection

The IoMT is a combination of a huge number of heterogeneous devices, networks, applications, and paradigms for data transfer that have distinct features and requirements. The detection of malware in a such complex environment is a very challenging task. The use of an "electronic health recorder (EHR)" to store the information of users over the IoT-based cloud for processing is one way to resolve heterogeneity issues in the IoMT.

Moreover, the BAN also produces a enormous amount of data in the IoMT environment. Therefore, in order to detect malware in a such complex environment, there is a need to develop a more specific method. Hence, more research is needed.

### 6.2. Cross-Platform Malware Detection

The deployment of malware detection tools in the IoMT environment is difficult because of its heterogeneity. It becomes tough to design such tools and mechanisms that could detect malware in a cross-platform environment. For instance, if there is an exchange of data and information from smart homes to smart healthcare to monitor the patients, the system becomes more convoluted, and, therefore, the detection of malware becomes a tedious task. Therefore, uninterrupted network-based malware techniques need to be developed.

### 6.3. Security Assessment

The methods of performing security research are versatile in their parameters. Different research groups or individuals carrying out research on the IoMT use different parameters for the analysis of security. Hence, there is no uniformity in the research. The use of adversarial analysis concepts and tools used by researchers does not consider the same parameters and inputs between different researchers. Therefore, comparing the proposed work of each individual or research group is quite challenging. Consequently, future research is required to define a set of parameters that could be used for research to strengthen the security of the IoMT.

### 6.4. Paradigm Shift in IoMT Sensors

The transition in society occurs at a faster pace when the new technologies emerge at a faster rate. The shifting from traditional to newer technologies, such as the IoMT, is the result of such a transition. A transition is currently taking place, from a generic method of tracking patients' health to clinical data integration, where sensors play a vital role in IoMT-based systems. Mobility in the IoMT is the principal feature that has revolutionized the tracking and caring of patients' health.

### 6.5. Security and Privacy

A significant amount of private and sensitive data is present in IoMT-based systems, and, over time, this considerable volume of data is increasing. The use of blockchain technology and cryptography are two vital elements that could make an IoMT-based system safer. The use of SHA-256 makes data integrity more efficient. The use of blockchain has entered into various fields of technology, and its use in IoMT would be very useful to achieve the integrity of data.

### 6.6. Blockchain for Healthcare Data Sharing

The data used in the IoMT-based system are highly sensitive. Apart from the healthcare employees, the patients are the complete controllers of these data. Therefore, there is always a chance of the leakage of data. The use of a time-stamped feature of blockchain could be very useful in detecting the integrity of such data and information. Once these data are stored in a distributed ledger, the detection of an anomaly in patients' data could be tackled efficiently.

### 6.7. Heterogeneity in an IoMT Communication Environment

The IoMT consists of a huge range of distinct devices and systems, such as full-edged systems, workstations, tablets, smartphones, RFID tags, etc. Communication occurs between these devices using an inter-connected network, and these devices use various protocols to communicate among themselves. These devices have their own differing storage capability, range of communication, and operating system, as well as power con-

sumption rate. Therefore, the detection of attacks in such a system needs to develop in a more specialized manner.

## 7. Research Challenges and Lessons Learned

The IoMT is evolving continuously, with the development of more digital medical devices and the advancement of new network security devices. Future research needs to focus on the below discussed points.

- The research on the network is very important for the IoMT. The backbone of the IoMT is the network through which each device communicates with others to yield the desired task. Therefore, if the device has to operate in a real-time scenario, as in the case of tele-surgery, the delay in transmission could result in the loss of human life. Hence, latency is one of the main concerns in such a scenario, where the healthcare worker is performing surgery through a haptic arm. Moreover, channel-based attacks are very deadly, as they could compromise the data on transit, and, hence, the integrity of the original data could be compromised. Therefore, it is crucial to deal with zero-day-based attacks over the communicating medium in the IoMT.
- The next area of concern for the future is the devices that are a part of the IoMT system. As we know that there is no standard architecture to be followed by everyone implementing IoT applications, so the same goes for the IoMT. There are devices, such as CCTV cameras and filed sensors, that do not have the capacity to update their software, and therefore become obsolete in terms of security after a particular time. Consequently, it is necessary to replace them with more advanced and newer versions. Hence, there could be future research on updating these devices while connected to the IoMT, rather than replacing them, in order to reduce the cost of implementing visual surveillance within the IoMT.
- The next area of research, which plays an important role in the function of the IoMT, is the application software and system software, which work, either as intermediates between the hardware and software or on top of them, for various medical processes. Some of the most widely used software in medical environments are electronic health records (EHR), hospital management systems (HMS), telemedicine software, etc. It is important to identify, on a regular basis, any vulnerabilities in the codes of these software, and there is therefore a need for a proper standardized framework to define the security check in this software. Moreover, there is a need to inspect the operating system code involved in the devices which are involved in the IoMT, in order to identify the existence of any zero-day attacks.
- The use of secure communicating channels, identification of vulnerabilities in the system at the right time, and the use of appropriate software and hardware to protect the IoMT application would not be possible without having effective governance, risk, and compliance (GRC) processes. These policies play a crucial role in the efficient working of the organization. Since, in the case of the IoMT, various more personal and sensitive data are involved, both locally and also remotely, it become very important to understand which of the patient's data require explicit permission from the patient for access, so that his/her fundamental rights are not violated. Furthermore, the policy should clearly identify the authorization domain for each employee, in order to safeguard against data breaches that may occur due to weak policy. Therefore, the use of efficient policy specifically for the IoMT is needed in future.

## 8. Conclusions

IoT-based connected medical and e-healthcare solutions, such as the IoMT, are proving to be revolutionary for the healthcare industry. The growth in related applications has been of immense help to the healthcare providers, such as clinics, hospitals, practitioners, and care givers, in providing patients with the most accurate, predictive, and effective medication strategies. Therefore, IoT-based solutions for e-healthcare services are emerging as a vital component to help match the demanding needs of the digital society. The IoMT

assists in real-time patient monitoring and allows all the related domains of healthcare to work together as one unit, with the help of an interconnected network. Furthermore, to maintain a high level of privacy, security, accuracy, and trust, it is highly recommended and essential to train IT and medical staff to ensure that they do not fall victim to cyber-attacks. Therefore, the main aim of this paper is to tighten the ties between the varied non-technical and technical solutions to enable increasingly efficient, secure, and sophisticated IoMT systems. In this paper, we discuss the security requirements, design challenges, novel attacks, and state-of-the-art security techniques that are essential to making IoMT-based systems more secure. The paper explores three vital security techniques required for securing IoMT systems, namely, asymmetric key algorithms, symmetric key algorithms, and keyless algorithms. Finally, the paper discusses various future research directions that will guide future researchers working in this area.

**Author Contributions:** Conceptualization, B.B. and A.K. (Avinash Kumar); methodology, A.K.A., P.B. and A.K. (Avinash Kumar); formal analysis, A.K. (Arun Kumar), A.K. (Amit Kumar) and A.K. (Avinash Kumar); investigation, P.B., A.K. (Amit Kumar) and A.K.A.; resources, A.K. (Avinash Kumar), B.B. and P.B.; writing—original draft preparation, A.K. (Avinash Kumar), A.K. (Arun Kumar) and B.B.; writing—review and editing, A.K.A., P.B. and A.K. (Amit Kumar); supervision, B.B.; funding acquisition, B.B. All authors have read and agreed to the published version of the manuscript.

**Funding:** This research received no external funding.

**Institutional Review Board Statement:** Not applicable.

**Informed Consent Statement:** Not applicable.

**Data Availability Statement:** Not applicable.

**Conflicts of Interest:** The authors declare no conflict of interest.

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
