# Peer review of "Towards a Secure and Sustainable Internet of Medical Things (IoMT): Requirements, Design Challenges, Security Techniques, and Future Trends"

_sustainability, doi:10.3390/su15076177_

Round 1

Reviewer 1 Report

1) The abstract is poorly written. The abstract should not be written as an introduction. It must provide a brief overview of the paper.

2) The Internet of things is not a new paradigm.

3) The paper does not analyze telesurgery or remote surgery's state-of-the-art strategy and suitable equipment (sensors and devices).

Author Response

1) The abstract is poorly written. The abstract should not be written as an introduction. It must provide a brief overview of the paper.

Thank you for your suggestions. We have rewritten the abstract and provided a brief overview of the paper as per comments.

2) The Internet of things is not a new paradigm.

We agree to this point and have revised the contents accordingly. 

3) The paper does not analyze telesurgery or remote surgery's state-of-the-art strategy and suitable equipment (sensors and devices).

Thank you for your suggestion. We agree that, telesurgery is a needed topic and must be discussed in this work. We have added a separate section highlighting the state-of-the-art strategy for telesurgery or remote surgery. Please check section 2.4, Page number 7.

Reviewer 2 Report

The article presents a review of a highly relevant topic, with an adequate number of references and reviewing important aspects of IoMT such as security, design challenges, and future trends. The article is very well organized, well written. I recommend its publication, however, there are some details that could be improved which are presented to the authors.

Author Response

The article presents a review of a highly relevant topic, with an adequate number of references and reviewing important aspects of IoMT such as security, design challenges, and future trends. The article is very well organized, well written. I recommend its publication, however, there are some details that could be improved which are presented to the authors.

Thank you for the appreciation. We have further modified the manuscript by adding 4 new sections.

Please check section 2.4 (State-of-the-art strategy for telesurgery or remote surgery) on page number 7.

Please check section 3.4 (Existing security frameworks for IoMT based applications) on page number 10 and 11.

Please check section 3.5 (Risk analysis and threat mapping) on page number 11 and 12.

Please check section 7 (Research challenges and lessons learned) on page number 20 and 21.

We have added some more recent references and compared the existing surveys and their contribution. Please check references [18-25] on page number 2.

Reviewer 3 Report

Dear Authors,

Please clarify the following concerns.

1. Mention the research gaps. (There are no research questions.)

2. Compare your study with recently published papers to show your paper's uniqueness. (What does it add to the subject area compared with other published material?)

3. The citations are missing in Figures 1 and 2. Also, figure 4 is typed as figure 3. The quality of the figures is poor.

4. The readability of the paper is poor.

Author Response

1. Mention the research gaps. (There are no research questions.)

Thank you for your suggestions. We have highlighted the research gaps in section 1. We have added some recent survey articles in this area and presented their contribution. Please check section 1, page number 2, references [18-25].

2. Compare your study with recently published papers to show your paper's uniqueness. (What does it add to the subject area compared with other published material?)

The required modifications are done as per review comments. Please check section 1, page number 2, references [18-25].

3. The citations are missing in Figures 1 and 2. Also, figure 4 is typed as figure 3. The quality of the figures is poor.

Thank you for your suggestions. We have cited all the figures properly and also corrected the figure numbering as per suggestions.

Figure 1-cited in line number 131

Figure 2-cited in line number 209

4. The readability of the paper is poor.

The entire manuscript is revised in order to enhance the overall readability of the paper. We have proof read the paper and corrected the grammatical errors and sentence formation issues. Further, some more new sections are added to enhance the overall quality.

Please check section 2.4 (State-of-the-art strategy for telesurgery or remote surgery) on page number 7.
Please check section 3.4 (Existing security frameworks for IoMT based applications) on page number 10 and 11.
Please check section 3.5 (Risk analysis and threat mapping) on page number 11 and 12.
Please check section 7 (Research challenges and lessons learned) on page number 20 and 21.
We have added some more recent references and compared the existing surveys and their contribution. Please check references [18-25] on page number 2.

Reviewer 4 Report

Summary: This paper highlights the significance of implementing suitable security measures and essential aspects of IoMT that makes it helpful in interconnecting various internal and external working domains of healthcare. Further, the paper explores various security requirements, inherent design challenges, and security techniques that could make IoMT more secure and sustainable. The following concerns should be taken care of for its improvement.

Comments:

1.       Authors should explicitly specify the novelty of their survey work. What progress against the most recent state-of-the-art similar surveys was made?

2.       Conclusions should be amended to incorporate a broader discussion of this specific study's significance and potential application.

3.       The author should present some existing frameworks/models on security and privacy Techniques in the Internet of Medical Things (IoMT) Environments (for ex. 10.1109/JBHI.2022.3187037, 10.1109/TNSM.2020.3007535) and compare them with various factors such as performance, efficiency, accuracy, ease of implementation, etc.

4.       Threat identification and risk analysis are equally crucial as mitigations. It's better to give an insight into recent advancements in Threat identification and risk analysis in IoMT environments.

5.       The author should add a section- Research challenges and lessons learned to cover the deep inside discussion on future and current research challenges and key lessons learned from a systematic investigation in Secure and Sustainable IoMT.

6.       English throughout the manuscript needs to be improved.

Author Response

Summary: This paper highlights the significance of implementing suitable security measures and essential aspects of IoMT that makes it helpful in interconnecting various internal and external working domains of healthcare. Further, the paper explores various security requirements, inherent design challenges, and security techniques that could make IoMT more secure and sustainable. The following concerns should be taken care of for its improvement.

Thank you for the appreciation and constructive comments. We have taken care of all the points in the revised version. The changes made are highlighted in yellow.

Comments:

1. Authors should explicitly specify the novelty of their survey work. What progress against the most recent state-of-the-art similar surveys was made?

Thank you for the suggestion. We have outlined the contributions of some recent surveys in this area and finally discussed the need of our survey. Please check section 1, references [18] to [25], page number 2.

2. Conclusions should be amended to incorporate a broader discussion of this specific study's significance and potential application.

Conclusion is modified and we have added a separate section related to lessons learned and research challenges.

3. The author should present some existing frameworks/models on security and privacy Techniques in the Internet of Medical Things (IoMT) Environments (for ex. 10.1109/JBHI.2022.3187037, 10.1109/TNSM.2020.3007535) and compare them with various factors such as performance, efficiency, accuracy, ease of implementation, etc.Thank you for your suggestion. We have discussed this in a separate section. Please check section 3.4 (Existing security frameworks for IoMT based applications) on page number 10 and 11. More existing frameworks have been discussed. Please check references [66] to [72].

4. Threat identification and risk analysis are equally crucial as mitigations. It's better to give an insight into recent advancements in Threat identification and risk analysis in IoMT environments.We have added a new section as per the comments. Please check section 3.5 (Risk analysis and threat mapping) on page number 11 and 12.

5. The author should add a section- Research challenges and lessons learned to cover the deep inside discussion on future and current research challenges and key lessons learned from a systematic investigation in Secure and Sustainable IoMT.Thank you for the suggestions. We have added a new section that focus on the research challenges and lessons learned. Please check section 7 (Research challenges and lessons learned) on page number 20 and 21.

6. English throughout the manuscript needs to be improved.

The entire manuscript is revised in order to enhance the overall readability of the paper. We have proof read the paper and corrected the grammatical errors and sentence formation issues. 

Round 2

Reviewer 3 Report

This paper can be accepted in its present form.

Author Response

Thank you for your constructive comments.

Reviewer 4 Report

The authors addressed all raised comments. The paper can be accepted in its present form.

Author Response

Thank you for your constructive comments and acceptance.